# Embryonic Stem Cells Can Generate Oral Epithelia under Matrix Instruction

**DOI:** 10.3390/ijms24097694

**Published:** 2023-04-22

**Authors:** Ridhima Das, Lisa Harper, Kayoko Kitajima, Tarig Al-Hadi Osman, Mihaela Roxana Cimpan, Anne Chr. Johannssen, Salwa Suliman, Ian C. Mackenzie, Daniela-Elena Costea

**Affiliations:** 1Gade Laboratory for Pathology and Center for Cancer Biomarkers CCBIO, Institute for Clinical Medicine, University of Bergen, 5020 Bergen, Norway; 2Institute for Cell and Molecular Science, Queen Mary University of London, London E1 4NS, UK; 3Department of Endodontics, The Nippon Dental University School of Life Dentistry at Niigata, Niigata 951-8580, Japan; 4Department of Clinical Dentistry, University of Bergen, 5020 Bergen, Norway; 5Department of Pathology, Haukeland University Hospital, 5021 Bergen, Norway

**Keywords:** mouse, embryonic stem cells, differentiation, skin, oral mucosa, matrix, fibroblast, niche

## Abstract

We aimed to investigate whether molecular clues from the extracellular matrix (ECM) can induce oral epithelial differentiation of pluripotent stem cells. Mouse embryonic stem cells (ESC) of the feeder-independent cell line E14 were used as a model for pluripotent stem cells. They were first grown in 2D on various matrices in media containing vitamin C and without leukemia inhibitory factor (LIF). Matrices investigated were gelatin, laminin, and extracellular matrices (ECM) synthesized by primary normal oral fibroblasts and keratinocytes in culture. Differentiation into epithelial lineages was assessed by light microscopy, immunocytochemistry, and flow cytometry for cytokeratins and stem cell markers. ESC grown in 2D on various matrices were afterwards grown in 3D organotypic cultures with or without oral fibroblasts in the collagen matrix and examined histologically and by immunohistochemistry for epithelial (keratin pairs 1/10 and 4/13 to distinguish epidermal from oral epithelia and keratins 8,18,19 to phenotype simple epithelia) and mesenchymal (vimentin) phenotypes. ECM synthesized by either oral fibroblasts or keratinocytes was able to induce, in 2D cultures, the expression of cytokeratins of the stratified epithelial phenotype. When grown in 3D, all ESC developed into two morphologically distinct cell populations on collagen gels: (i) epithelial-like cells organized in islands with occasional cyst- or duct-like structures and (ii) spindle-shaped cells suggestive of mesenchymal differentiation. The 3D culture on oral fibroblast-populated collagen matrices was necessary for further differentiation into oral epithelia. Only ESC initially grown on 2D keratinocyte or fibroblast-synthesized matrices reached full epithelial maturation. In conclusion, ESC can generate oral epithelia under matrix instruction.

## 1. Introduction

Cases of extensive oral mucosa defects after trauma or cancer surgeries are devastating conditions and have high therapeutical demands [1]. Similarly, severe burns or toxic epidermal necrolysis with various degrees of skin loss or detachment pose major reconstructive challenges [2]. Currently, the clinical ‘gold standard’ treatment in such injuries is split-thickness autologous skin grafting [3,4,5]. Firstly, skin and then mucosal sheets generated in vitro and transplanted in vivo have emerged as an attractive avenue for skin and mucosal regeneration, and significant progress has been made over the years in the development and clinical use of such bioengineered products [6]. However, these are not implemented in clinical practice due to lack of commercially available tissue-engineered skin or mucosal replacements that can fully replace the functional and anatomical properties, although they are cost-effective, readily available, user-friendly, and have long shelf life [7,8]. The generation of fully functional skin and oral mucosal sheets is today restrained by our limited understanding of the normal differentiation process of epithelium and how we can use alternative stem cell sources for epithelial regeneration. 

Undergoing research on various types of stem cells, such as epidermal, dermal, and mesenchymal stem cells from bone marrow in adult tissues, and pluripotent embryonic stem cells (ESC) as sources or catalyzators for complete regeneration of skin and oral mucosa is bringing promise to understanding and recapitulating the functional repair and regeneration of skin and oral mucosa [9,10,11]. Pluripotent stem cells, whether embryonic (ESC) or induced (iPS) by reprograming of adult somatic cells, are defined by their capacity of self-renewal and multilineage differentiation. They have the potential to generate indefinite numbers of distinctive cell types in vitro, which makes them useful for understanding the mechanisms involved in the development of various cell and tissue types [12]. Due to ethical issues, research on human ESC is limited, but mouse ESC represent an important tool to study mechanisms of differentiation into multiple cell types and the development of various tissues [13]. Successful culture of mouse ESC, i.e., pluripotent cells derived from the inner cell mass of the blastocyst, was first reported in 1981 [14,15]. Essential for preserving the pluripotency of mouse ESC in culture is the leukemia inhibitory factor (LIF), a member of the cytokine family interleukin-6 (IL-6), which is an activator of the STAT3 transcription factor that controls stem/progenitor cells’ renewal and differentiation [16,17,18]. 

During embryonic development, epithelium forms as a result of reciprocal interactions between mesoderm and ectoderm or endoderm. Previous studies have indicated that ESC can be induced into epithelial cells of stratified epithelium (keratinocytes) when seeded on extracellular matrix (ECM) in the presence of bone morphogenic protein-4 (BMP-4) or vitamin C [19]. ECM is a complex biological scaffold that provides a favorable microenvironment for growth of different cells, in addition to providing functional and structural support [20]. ECM regulates various cell functions, such as proliferation, migration, and differentiation, as well as cell survival [21]. Most known naturally derived ECMs are collagen, laminin, fibrin, and Matrigel, which are regularly used for in vitro expansion of different cell types [22]. The interactions between stem cells and the supporting ECM play a crucial role in differentiation of these cells [23]. The ECM mimicking the stem cell niche in vitro helps in guiding stem cells to a lineage-specific cell, which represents a milestone in the field of tissue engineering [24].

The aim of this study was to find out whether cues from ECM can induce differentiation of uncommitted pluripotent stem cells into an epithelium of a particular regional phenotype, for example, skin or oral mucosa, and for this, we used mouse ESC as a model of pluripotent stem cells. We hypothesized that uncommitted ESC can be induced into epithelial lineages by exposure to the matrix products of mesenchymal cells and that epithelia derived from such cells would develop phenotypic characteristics related to the matrix components they were exposed to (Appendix A Figure A1). 

## 2. Results

### 2.1. Vitamin C Alone Initiated Differentiation of ESC towards the Epithelial Lineage

Under standard undifferentiating conditions of medium containing LIF and growing on gelatin, ESC grown in 2D showed the formation of tight colonies as normally formed by ESC with a high expression of SSEA-1 (38.5 ± 3.2%) (Figure 1A) and no expression of PanCK (less than 1%) (Figure 2A). When LIF was removed, ESC became scattered and flattened and acquired a differentiated morphology, with very few tight colonies and loss of ESC markers SSEA-1 and OCT-3/4 expression. However, lack of staining with the PanCK antibody indicated no differentiation into epithelial phenotypes. Further loss of tight colonies with cell scattering and flattening, with a further decrease of SSEA-1 expression till 15.5 ± 2.4%, was associated with the addition of vitamin C but not BMP-4. The percentage of ESA cells showing PanCK expression was less than 1% when grown in the presence of BMP-4 but rose to 1.7 ± 0.4% with vitamin C. 

### 2.2. ECM Generated by Both Fibroblasts and Keratinocytes Enhanced Differentiation of ESC into Epithelial Lineages

Plating ESC in 2D on all types of matrices investigated further enhanced their differentiation towards an epithelial phenotype, as observed by light microscopy (Figure 1D–H) and by immunofluorescence for visualization of PanCK (Figure 2A–H). Quantification by FACS analysis showed that PanCK expression significantly increased to 7.0 ± 0.6% (*p* < 0.05%) in ESC grown on laminin, to 5.3 ± 0.7% (*p* < 0.05) when ESC were grown on oral fibroblast-derived matrix, and to 8.6 ± 1.3% (*p* < 0.05) when ESC were grown on oral keratinocyte-derived matrix. Control keratinocytes derived from oral and ear skin showed high expression (98–100%) of PanCK and low expression (1.7 ± 0.5%) of SSEA-1 stem cell marker.

### 2.3. Three-Dimensional Culture on Simple Collagen Induced Further Differentiation of ESC in Epithelial and Mesenchymal Lineages but without Formation of a Lining Epithelium

Both non-primed and 2D matrix-primed ESC developed into two morphologically distinct cell populations when grown for 14 days in 3D organotypic cultures on simple collagen gels (Figure 3, Figure 4 and Figure 5): (i) epithelial-like cells with dense nuclei and eosinophilic cytoplasm organized in islands scattered in an eosinophilic matrix with occasional cyst- or duct-like structures (Figure 3), all of which were PanCK positive (Figure 5B) and (ii) spindle-shaped cells with little cytoplasm and elongated nuclei (Figure 3), which were vimentin positive (Figure 5C), suggestive of mesenchymal differentiation, a few of which migrated into collagen matrix. The histological picture did not change for the 3D cultures of the non-primed ESC grown on top of simple collagen gels (Figure 3A,B) after one more week in 3D cultures (Figure 4A,B). 

### 2.4. Two-Dimensional Priming on Oral Fibroblast-Produced ECM and Further 3D Culture on Collagen Induced Differentiation of ESC into a Lining Epithelium

When ESC that were first primed in 2D on oral fibroblast-produced ECM (OF-ECM) were seeded on top of collagen gels, they formed a lining epithelium of 1–2 cell layers in thickness (Figure 3E). Notably, formation of cyst-like structures with a central lumen and a 1–2 cell layer epithelial lining was observed when OF-ECM-primed ESA were cultured in 3D on simple collagen gels. The impression was that the surface epithelium was formed through a process of ‘opening’ towards the surface of multiple such cyst-like structures. After one more week in 3D culture, fewer cyst-like structures were observed in these cultures and the presence of almost a continuous superficial lining of 1–2 cell layers was observed (Figure 4E), with focal stratification (Figure 4F). In some areas, this epithelium took the appearance of a pseudostratified columnar epithelium (Figure 5D), reminiscent of respiratory epithelium. Epithelium of these cultures was positive for simple epithelial keratin CK19 (Figure 5E). 

### 2.5. Two-Dimensional Priming on Oral Keratinocyte-Produced ECM (OK-ECM) and Further 3D Culture on Collagen Induced Differentiation of ESC into Ductal and Mucus-Producing Epithelium

When ESA were first primed on OK-ECM and then further cultured in 3D on top of simple collagen gels for 14 days, in addition to solid epithelial sheets and ductal structures, scattered single mucin-producing cells (green arrow and circle in Figure 3I,J) were observed. Numerous mucin-producing cells were observed one week later (Figure 4I,J). These 3D cultures contained numerous cyst-like structures containing a mixture of duct-like cells, mucin-producing cells, and keratinized cells. These cysts failed to ‘open’ on the surface to give rise to a lining epithelium, unlike the cyst-like structures observed in cultures with the ESA primed on OF-ECM that ‘opened’ and formed a lining epithelium when given more time to mature. 

### 2.6. Full Epithelial Phenotypic Specificity Was Achieved after 2D Priming on Matrix Products of Regionally Distinct Fibroblasts and Keratinocytes and Further 3D Culture on Fibroblast-Populated Collagen Gels

When the non-primed ESC were cultured in 3D on oral fibroblast containing collagen matrix for 14 days, only an increase in the number of duct-like structures was observed but no formation of lining epithelium, not even after one more week in culture (Figure 3C). When OF-ECM-primed ESA cells were grown on oral fibroblast-populated collagen gels (Figure 3G,H), formation of a squamous stratified multilayered epithelium, similar to stratified oral non-keratinized epithelium, was observed, mainly lining the cyst-like structures. After 7 more days, very few of these cyst-like structures were still present, and a superficial, almost continuous layer composed of a stratified squamous epithelium on top of collagen gels was observed (Figure 4G,H), with focal areas of parakeratinization (Figure 5F) and positive for CK 13 (Figure 5G).

Of interest, no mucus-producing cells were observed when OK-ECM-primed ESA cells were grown in 3D cultures on top of oral fibroblast-containing gels (Figure 3K,L and Figure 4K,L). In these cultures, the presence of an almost continuous superficial epithelial lining of 1–2 cell layers was observed already after 14 days in 3D cultures (Figure 3K,L), which became an almost continuous stratified squamous epithelium after one more week, with focal areas of keratinization (Figure 4K,L). When OK-ECM-primed ESA cells were grown in 3D cultures on top of skin fibroblast-containing gels, focal areas of full differentiation into orthokeratinized stratified squamous epithelium on top of 3D cultures were observed, and these were positive for CK 10 (Figure 5H,I).

## 3. Discussion

We show in this study that vitamin C enhanced the number of cells expressing epithelial markers, but matrix synthesized by either oral/skin fibroblasts or keratinocytes was required to induce the expression of a stratified epithelial phenotype. Vitamin C plays a critical role in regeneration and wound healing [25]. It regulates the differentiation of keratinocytes by regulating the function of AP-1 complexes [26]. Guenou et al. (2009) have shown that hESC induced with vitamin C and BMP-4 were cytokeratin 14 positive [27]. It is a well-known fact that LIF plays a critical role in maintaining the pluripotency of the embryonic stem cells in vitro [28,29]. This study shows that with LIF removal in both 2D and 3D cultures, ESC underwent differentiation.

Coraux et al. suggested already in 2003 that once primed, ESC can differentiate into epithelial tissues, given sufficient time and continuous instruction from vital fibroblasts [19]. However, to our knowledge, their interesting work has not been replicated by others. Our findings presented here support their suggestion. In view of reports suggesting that epithelial differentiation of stem cells occurs within an epithelial context, we also included an additional condition of ESC growth on matrix produced by keratinocytes [30], and our study shows, in addition to the data on fibroblast-derived matrix, that also keratinocyte-produced matrix and laminin enhance the induction of a keratinocyte phenotype. Of interest, this was the only condition that generated mucus-producing cells, but it was insufficient to generate a superficial epithelium. Further 3D culture with vital fibroblast instruction for 15 days was necessary for further differentiation into mature, regionally relevant epithelial structures. We used only collagen gels for reconstructing the connective tissue equivalent of the 3D cultures. Several other substrates have been shown to support formation of a fully differentiated keratinized gingival mucosa when using adult somatic human oral cells [31]. An obvious further step of this study would be to test these substrates for engineering fully differentiated (keratinized) gingival mucosa from mouse ESC or from human induced pluripotent stem cells.

We showed here, by staining with different keratin antibodies, that a specific keratin pattern can be induced in ESC grown on an extracellular substrate, depending on the region of origin of the cells that generated that substrate. The differences in tissue patterns observed indicate that this may be valuable for generating epithelial tissues with defined regional specificities. This is to be expected since ESC have been shown to have the ability to differentiate into any type of cell when provided proper induction in vitro [32,33,34]. This characteristic ability of ESC has been exploited in regenerative medicine for more than a decade now; however, the use of hECS is prevented by ethical considerations. We therefore used mouse ESC, and our results indicate that pluripotent stem cells can be a promising source of skin/mucosal sheets, which can be used in regenerative medicine and clinics [35], and that regional fibroblasts are essential for full epithelial maturation, provided that a clinical-grade xeno-free protocol would be developed in the future. 

## 4. Materials and Methods

### 4.1. Cell Culture

Mouse ESC of the feeder-independent cell line E14.2 were routinely cultured on 0.1% gelatin (Gibco, New York, NY, USA) in 0.1 M of phosphate-buffered-saline (pH 7.2) (Gibco)-coated plates and grown in a medium consisting of Dulbecco’s modified Eagle medium (DMEM-D6429, Sigma, Welwyn Garden City, UK) supplemented with 20% of fetal calf serum (FCS-11573397, Fisher Scientific, Leicestershire, UK), 1% non-essential amino acids (11350912, Fisher Scientific), 1% glutamine (Gibco), 1% penicillin/streptomycin (Gibco), 0.1% mercaptoethanol, and 0.1% LIF (Gibco) [36,37,38]. 

Primary keratinocytes and fibroblasts were isolated from murine skin (ear) and oral mucosa (buccal) as previously described (*n* = 5 C57BL/6 mice, Jackson Laboratory, Bar Harbor, ME, USA) [39]. Keratinocytes were routinely grown on plastic surfaces (Nunc, Naperville, IL, USA) with no feeding layers, in keratinocyte serum-free medium (KSFM) (Gibco, Grand Island, NY, USA) supplemented with 1 ng/mL of human recombinant epidermal growth factor (Gibco), 25 μg/mL bovine pituitary extract (Gibco), 2 mM L-glutamine (Gibco), 100 U/mL penicillin (Gibco), 100 μg/mL streptomycin (Gibco), and 0.25 μg/mL amphotericin B (Gibco). Fibroblasts were grown in DMEM as described above. The study was approved by the National Animal Research Authority (FOTS ID 2006400).

### 4.2. Culture Conditions for 2D Differentiation

Differentiation of ESC was attempted in two steps. The first step was called ‘priming’, where ESC were grown (seeding density of 1 × 10^5^ cells/cm^2^) in 2D in different media conditions: with or without LIF, combined with 0.3 mM of vitamin C/L-ascorbic acid, with 0.5 nM of recombinant mouse BMP-4 protein (R&D Systems, Abingdon, UK), and on different matrices (Appendix A Table A1). In addition to routinely used gelatin for ESA culture, commercially available laminin (Cultrex 3-D Culture Matrix Laminin I, R&D systems) and various keratinocyte (oral and skin) and fibroblast (oral and skin)-derived ECM matrices were used. 

To obtain the ECM, cells were grown till confluence and then gently removed at room temperature using 20 mM of EDTA in Hank’s Balanced Salt Solution (Gibco) containing 20 mM of HEPES (Sigma). ESC were immediately seeded on freshly ECM-coated plates, and when this was not possible, the plates were stored in PBS at 4 °C until use for ESC culture. ESC (1 × 10^5^ cells/cm^2^) were cultured for 8 days on gelatin or ECM-coated plates and then removed from these matrices using trypsin, counted, and then further cultured in three-dimensional (3D) organotypic cultures.

### 4.3. Three-dimensional Organotypic Cultures

Simple collagen gels (700 μL for each culture) were prepared on ice by mixing 7 vol. (3.40 mg/mL) of rat tail collagen type I (Collaborative Biomedical, Bedford, MA, USA), 2 vol. reconstitution buffer (261 mM NaHCO3, 150 mM NaOH, 200 mM HEPES) with pH 8.15, 1 vol. DMEM 10× (Sigma), and 1 vol. FCS. ESC were removed from the matrices mentioned above by trypsin, counted, and then plated onto collagen gels containing either no cells, oral fibroblasts (OF, 0.5 × 10^6^ cells/mL of matrix), or skin fibroblasts (SF, 0.5 × 10^6^ cells/mL of matrix) and maintained for 7 days submerged in the 3D culture medium. This consisted of serum-free FAD medium (DMEM: Ham’s F-12/3:1) supplemented with 1 μM of hydrocortisone, 0.8 μM insulin, 0.25 mM transferrin, 0.3 mM L-ascorbic acid, 15–30 μM linoleic acid, 15 μM bovine serum albumin, and 2 mM L-glutamine (all from Sigma). Collagen gels were then lifted to air/medium interface for a further 7 or 14 days, and the resulting tissues were fixed, embedded in paraffin, sectioned, and stained for histological examination. Presence or absence of LIF and vitamin C, different matrix priming regimes, and collagen cell content resulted in 8 different experimental conditions for the 3D cultures and two time points, each repeated in triplicates.

### 4.4. Immunofluorescence

To quantify the extent of differentiation into epithelial lineages after ‘priming’ of ESC in 2D cultures, the expression of pancytokeratin (PanCK, DAKO A/S, Glostrup, Denmark, titration 1:200) as a marker of epithelial differentiation was analysed. Expression of embryonic stem cell markers, such as stage-specific embryonic antigen-1 (SSEA-1 or CD15, Clone MC-480, Stem Cell Technologies, Cambridge, UK) and OCT-3/4 (POU5F1, Clone 40, Stem Cell Technologies), was also investigated. Cells were grown on cover slips and fixed with 4% of paraformaldehyde (Sigma) for 15 min. Cells were then washed with PBS before being permeabilized using 0.3% of Triton for 10 min and then again washed with PBS. Blocking was done using 5% of BSA (Sigma) in PBS for 45 min, followed by primary antibody (1 ug/mL) for 60 min. Cells were then washed with PBS, and secondary antibody (goat anti-mouse IgG, FITC, Stem Cell technologies) was added for 60 min in the dark, which was followed by mounting in Mounting Medium with DAPI—Fluoroshield (Abcam, UK). The IgG1 isotype primary antibody was used for negative controls.

### 4.5. Flow Cytometry

Cells were trypsinized, washed in PBS, and placed in ice-cold PBS with 2% of FBS and 1% HEPES at a density of 5 × 10^6^ cells/500ul. The primary antibodies 1:20 Anti-SSEA1, 1:20 Anti-pancytokeratin (DAKO) and the isotope control were added and incubated for 45 min in the dark at 4 °C. After washing with PBS and centrifuging at 1500 rpm for 5 min at 4 °C, cells were incubated with Alexa Fluor 488, conjugated with anti-rabbit secondary antibody, and then analyzed by flow cytometry (Accuri™ C6, BD Biosciences, New Zealand). For each sample, 10,000 live particles were analyzed. Data analysis was performed using the Accuri™C6 software (Auckland, https://www.bdbiosciences.com/en-nz/products/instruments/flow-cytometers/research-cell-analyzers/bd-accuri-c6-plus, accessed on 10 January 2016). The data are presented as mean +/− SD. Statistical analysis was performed using the ANOVA test with a level of significance set at 5% (SPSS 11.0).

### 4.6. Immunohistochemistry

ESC ‘primed’ on 2D matrices were grown in the second step of differentiation in 3D organotypic cultures, and the tissues obtained were first examined histologically after hematoxylin and eosin (Sigma) staining. The regional specificity of the differentiated epithelium was further examined by immunohistochemistry: keratin pairs 1/10 (CK1/10 DAKO, titration 1:10) and 4/13 (CK4/13 Novocastra, New Castle, UK, titration 1:50) were used to distinguish keratinized (skin) versus non-keratinized (oral) epithelial phenotype; keratins 8,18,19 (CK8, CK18, CK19 from Cancer Research UK, 1:5 titration) were used to phenotype simple epithelia, and vimentin (titration 1:200, DAKO) was used as a marker of differentiation towards a mesenchymal phenotype [40]. The immunohistochemical staining was carried out using the DAKO Autostainer—Universal Staining System (DAKO). In total, 5 μm thick formalin-fixed paraffin-embedded 3D organotypic sections were used. All sections were processed as previously reported [40]. The presence of antigen was visualized with DAB+ (3,3′-diaminobenzidine, DAKO). Mouse skin and buccal mucosa were used as positive controls.

## Figures and Tables

**Figure 1 ijms-24-07694-f001:**
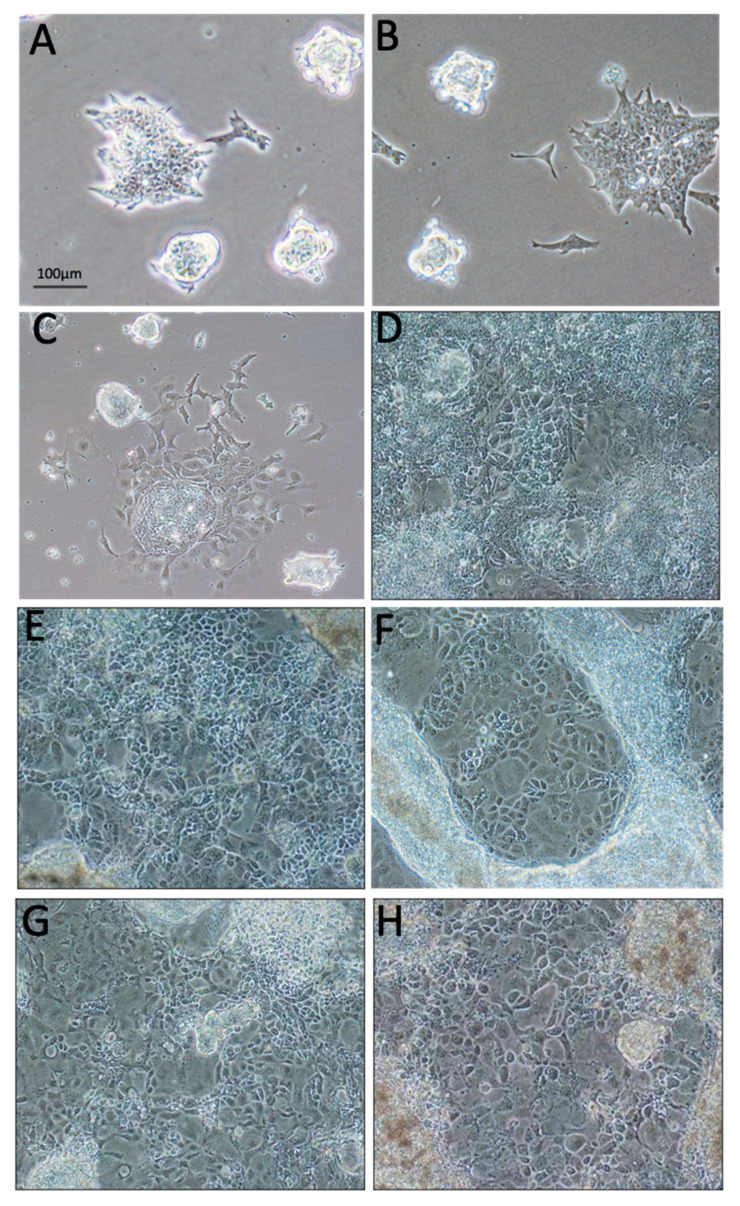
Representative light microscopy images of ESA cells grown in 2D in different media and substrates. (**A**) ESA grown on routine medium with LIF. (**B**) ESA cells grown in routine medium without LIF. (**C**) ESA cells grown in routine medium without LIF and with vitamin C. (**D**) ESA cells grown in routine medium without LIF and with vitamin C and plated on laminin-coated dishes. (**E**) ESA cells grown in routine medium without LIF and with vitamin C and plated on SK-ECM-coated dishes. (**F**) ESA cells grown in routine medium without LIF and with vitamin C and plated on OK-ECM-coated dishes. (**G**) ESA cells grown in routine medium without LIF and with vitamin C and plated on SK-ECM-coated dishes. (**H**) ESA cells grown in routine medium without LIF and with vitamin C and plated on OF-ECM-coated dishes.

**Figure 2 ijms-24-07694-f002:**
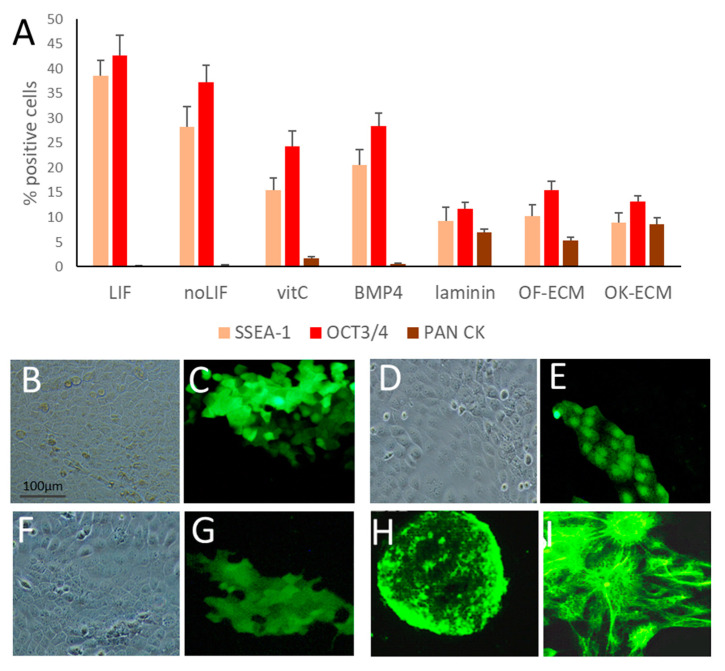
Flow cytometry and immunofluorescence images of ESC grown on different substrates. (**A**) Bar graph showing quantification of stem cell markers (SSEA-1 and OCT-3/4) and stratified squamous epithelial differentiation marker PanCK. (**B**,**C**) Light microscopy and immunofluorescent staining for squamous epithelial differentiation marker PanCK of a colony of ESC grown on laminin. (**D**,**E**) Light microscopy and immunofluorescent staining for PanCK of a colony of ESC grown on OF-ECM. (**F**,**G**). Light microscopy and immunofluorescent staining for PanCK of a colony of ESC grown on OK-ECM. (**H**) Immunofluorescent staining for SSEA-1 of a colony of ESC grown routinely in medium containing LIF and on gelatin. (**I**) Immunofluorescent staining for PanCK of a colony of oral keratinocyte cell line OKT6, the positive control for stratified squamous epithelial cells. Cells in (**B**–**G**) were all grown in growth medium without LIF and with vitamin C. Scale bar shown in (**B**) applies to all images.

**Figure 3 ijms-24-07694-f003:**
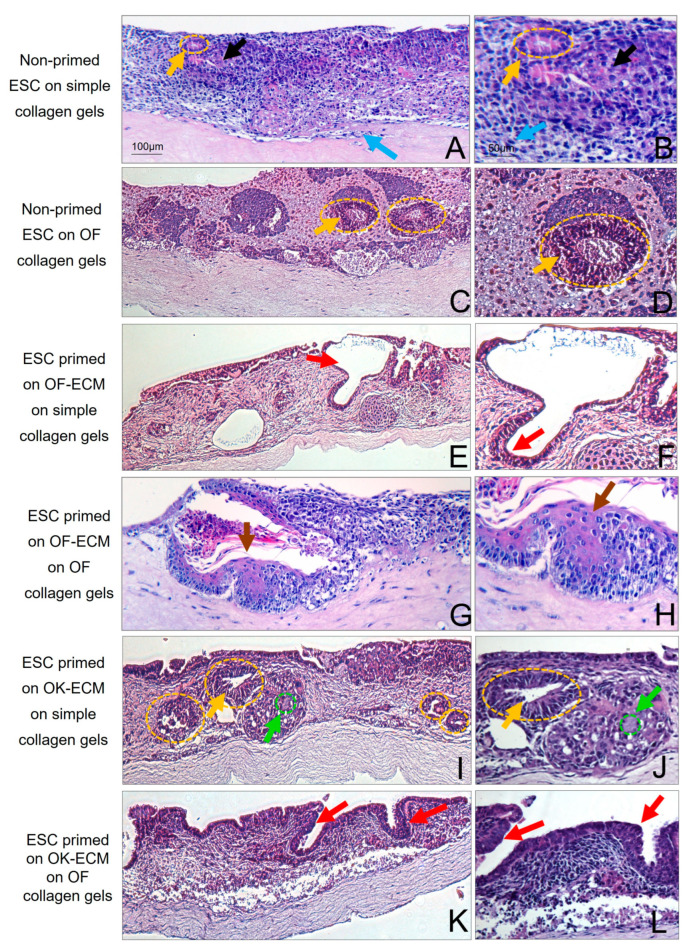
Histological images (hematoxylin-eosin) of 3D organotypic cultures of ESA cells firstly primed in 2D on different substrates, then seeded in 3D on top of various collagen matrices and grown for 14 days. Black arrows point towards cells with typical epithelial cell-like morphology with abundant eosinophilic cytoplasm, and blue arrows point towards cells with typical mesenchymal cell-like morphology with scant cytoplasm and elongated shape (**A**,**B**). Yellow arrows and circles point to epithelial duct-like structures (**A**–**D**,**I**,**J**). Red arrows point towards cyst-like epithelial lining (**E**,**F**,**K**,**L**), with occasional multilayered stratification (**K**,**L**). Brown arrows point towards stratified squamous keratinized-like epithelium (**G**,**H**). Green arrows and circles point towards mucin-like-producing cells (**I**,**J**). Scale bars shown in (**A**,**B**) apply to all images below in the columns.

**Figure 4 ijms-24-07694-f004:**
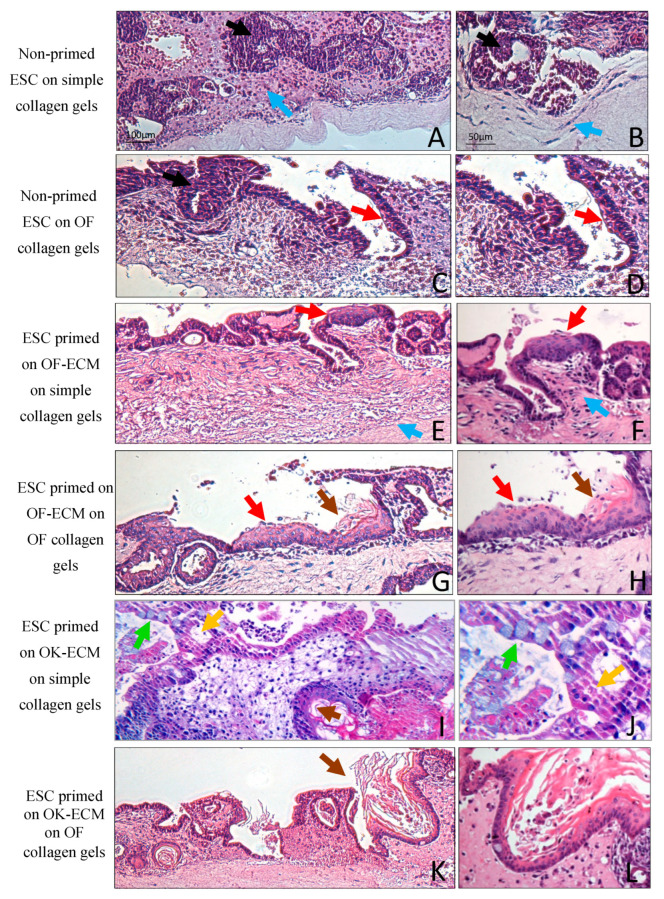
Histological images (hematoxylin-eosin) of 3D organotypic cultures of ESA cells firstly primed in 2D on different substrates, then seeded in 3D on top of various collagen matrices and grown for 21 days. Black arrows point towards cells with typical epithelial cell-like morphology with abundant eosinophilic cytoplasm (**A**–**C**), and blue arrows point towards cells with typical mesenchymal cell-like morphology with scant cytoplasm and elongated shape (**A**,**B**,**E**,**F**). Yellow arrows point to epithelial duct-like structures (**I**,**J**). Red arrows point towards cyst-like epithelial lining (**C**–**H**), with occasional multilayered squamous stratification (**E**–**H**). Brown arrows point towards stratified squamous keratinized-like epithelium (**G**–**I**,**K**,**L**). Green arrows point towards mucin-like-producing cells (**I**,**J**). Scale bars shown in (**A**,**B**) apply to all images below in the columns.

**Figure 5 ijms-24-07694-f005:**
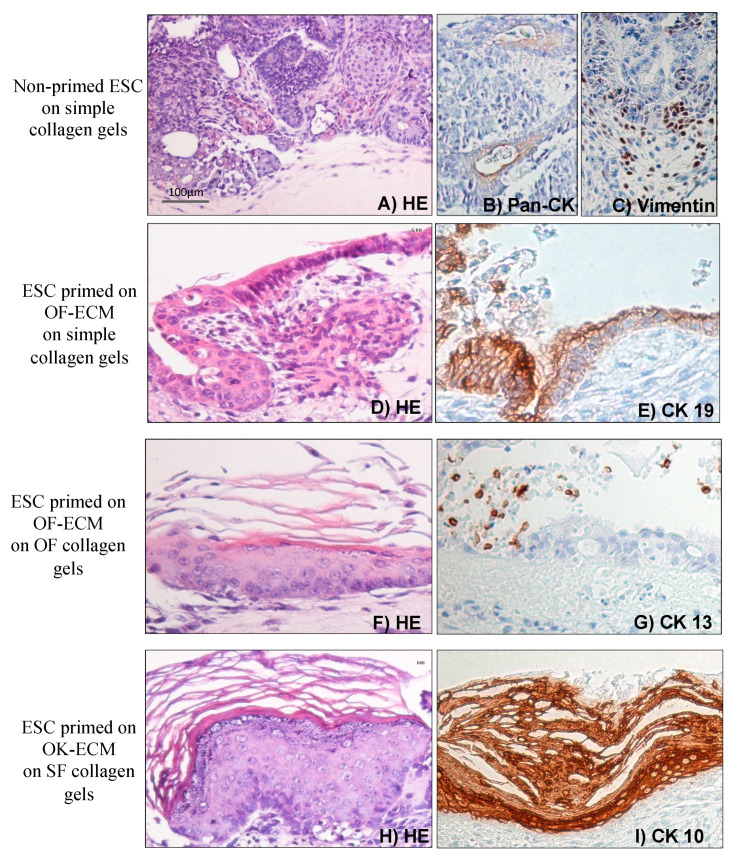
Representative microscopy images of 3D organotypic cultures of ESC firstly primed in 2D on different substrates, then seeded in 3D on top of various collagen matrices. (**A**–**C**) non-primed ESC grown in 3D on simple collagen gels. (**D**,**E**) ESC primed in 2D on OF-ECM, then grown in 3D on simple collagen gels. (**F**,**G**) ESC primed in 2D on OF-ECM, then grown in 3D on OF-populated collagen gels. (**H**,**I**) ESC primed in 2D on OK-ECM, then grown in 3D on SF-populated collagen gels. Scale bar shown in (**A**) applies to all images.

## Data Availability

The data presented in this study are available upon request from the corresponding author.

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
