# Peer review of "Embryonic Stem Cells Can Generate Oral Epithelia under Matrix Instruction"

_ijms, 2023, doi:10.3390/ijms24097694_

Round 1

Reviewer 1 Report

In this article, authors tried to address whether molecular cues from ECM can induce differentiation of uncommitted pluripotent stem cells using murine ESC as a model and showed that ECM instruction renders ESCs to differentiate into cells of epithelial lineage with phenotypic characteristic of related ECM e.g. Induction of keratinocyte differentiation by oral keratinocyte produced matrix. However, the molecular basis of this remains to be understood and was not addressed here.  

The authors showed that full epithelial phenotypic specificity was achieved after 2D priming on matrix products of regionally distinct fibroblasts and keratinocytes and further 3D culture on fibroblasts-populated collagen gels. This observation is supported in part by Fig 5 (H, I). However, the authors did not show or mention if the observation stands correct for the combination of OF-ECM primed ESCs cultured on skin fibroblast (SF) populated collagen gels.  

Further, any particular reason for not including ESCs primed on OK-ECM/OF-ECM and grown on SF populated collagen gels? 

It would be interesting to know the differentiation trajectory of ESCs primed with ECM of mixed origin, e.g. OF-ECM with OK-ECM or OF-ECM with SF-ECM or SF-ECM with OK-ECM.  

Formatting comments: 

  1. Figure legends could be made more concise by omitting repetitive sentences.   
  2. Matrix abbreviations are inconsistent between table 1 and the main text throughout the manuscript.  
  3. Gelatin is mistyped. 
  4. Line 118: text says skin fibroblast-derived matrix while figure 1 legend says oral keratinocyte-derived matrix, which one is correct?
  5. Figure 1: The culture conditions written on the bar diagram are confusing. As I understand, for differentiation, cells were cultured without LIF and in the presence of vitamin C or BMP4. For culturing cells on surfaces coated with laminin/OF-ECM/OK-ECM, growth medium without LIF and with vitamin C was used.  
  6. 2D and 3D texts are mentioned incorrectly at the beginning of paragraphs in the result section. 
  7. Label L is missing from figure 3, panel L. 
  8. Appendix figure A. what denotes Ex? 
  9. Line 79: I believe the authors meant cues instead of clues.  
  10. Line 127: Orthokeratinized is mistyped.  
  11. Line 282: degree C is incorrect. 
  12. Line 283: cell number is written incorrectly. 

Author Response

Thank you for the thorough review of our paper that helped improving it. Please find bellow point by point our answers. We have also revised the English language and we hope that the manuscript is now acceptable for publication.

Comment: In this article, authors tried to address whether molecular cues from ECM can induce differentiation of uncommitted pluripotent stem cells using murine ESC as a model and showed that ECM instruction renders ESCs to differentiate into cells of epithelial lineage with phenotypic characteristic of related ECM e.g., Induction of keratinocyte differentiation by oral keratinocyte produced matrix. However, the molecular basis of this remains to be understood and was not addressed here. 

Answer: The reviewer is correctly pointing that we did not reveal the molecular mechanisms behind the differentiation of ESCs towards epithelial lineage under the influence of ECM. This study aimed only at testing the hypothesis that ECM is able to induce differentiation of ESCs towards the epithelial lineage and shows results in this line. We are currently investigating this further and hope to come with a follow up manuscript pinpointing the more specific molecular clues.

Comment: The authors showed that full epithelial phenotypic specificity was achieved after 2D priming on matrix products of regionally distinct fibroblasts and keratinocytes and further 3D culture on fibroblasts-populated collagen gels. This observation is supported in part by Fig 5 (H, I). However, the authors did not show or mention if the observation stands correct for the combination of OF-ECM primed ESCs cultured on skin fibroblast (SF) populated collagen gels. Further, any particular reason for not including ESCs primed on OK-ECM/OF-ECM and grown on SF populated collagen gels? It would be interesting to know the differentiation trajectory of ESCs primed with ECM of mixed origin, e.g. OF-ECM with OK-ECM or OF-ECM with SF-ECM or SF-ECM with OK-ECM. 

Answer:  We did not mention the results from the combination of OF-ECM primed ESCs cultured on skin fibroblast (SF) populated collagen gels because we did not include this combination in our experimental setup. We thank also the reviewer for the suggestion on testing the effect of ECM of mixed origin that we did not consider in this study, but we will include in the next set of experiments. 

Formatting Comments:

  1. Figure legends could be made more concise by omitting repetitive sentences.  Matrix abbreviations are inconsistent between table 1 and the main text throughout the manuscript. Gelatin is mistyped.

Answer: Thank you very much for these comments. We have amended them.

2. Line 118: text says skin fibroblast-derived matrix while figure 1 legend says oral keratinocyte-derived matrix, which one is correct?

Answer: Thank you very much for spotting this confusion. Correct is as in figure 1 legend: oral keratinocyte-derived matrix. The text has been now corrected.

3. Figure 1: The culture conditions written on the bar diagram are confusing. As I understand, for differentiation, cells were cultured without LIF and in the presence of vitamin C or BMP4. For culturing cells on surfaces coated with laminin/OF-ECM/OK-ECM, growth medium without LIF and with vitamin C was used. 

Answer: We think the reviewer meant here Figure 2. Yes, it is correct that the conditions in the bar diagram are as mentioned by the reviewer. We agree that it would be more readable to have more comprehensive description of the conditions in the graph, but it is difficult to write more on the X-axis of figure 2A.

  1. 2D and 3D texts are mentioned incorrectly at the beginning of paragraphs in the result section.

Answer: Thank you. This has been now corrected.

5. Label L is missing from figure 3, panel L.

Answer: Thank you. This has been now corrected.

6. Appendix figure A. what denotes Ex? epithelial cells

Answer: We apologize for omitting defining this abbreviation in the figure legends. E stands for ‘epithelial cells’ and x was used to denote any region in the body those cells come from. This has been now added to the figure legends.

7. Line 79: I believe the authors meant cues instead of clues. 

8. Line 127: Orthokeratinized is mistyped. 

9. Line 282: degree C is incorrect.

10. Line 283: cell number is written incorrectly.
Answer: Thank you for pointing these mistakes to us. They are now corrected.

Reviewer 2 Report

Das et al. investigated the effects of the ECM on the differentiation of pluripotent stem cells into oral epithelia. The novelty of this paper is low, because it is well known that ESC can differentiate with high efficiency in all the different types of cells and therefore are a promising source of skin/mucosa. 

1) Please add the statistical analysis performed in the material an method section

2) which is the seeded cell density in the 2D conditions? From the figure 1 it seems that it is very different among the experimental conditions, and it can affect also the expression of the pluripotent and epithelial differentiation markers, because when ESC overgrowth they start to differentiate spontaneusly. 

Author Response

Thank you for the thorough review of our paper that helped improving it. Please find bellow point by point our answers. We have also revised the English language and we hope that the manuscript is now acceptable for publication.

Comment: Please add the statistical analysis performed in the material and method section.

 Answer: Thank you for this suggestion. We have added this now in the material and methods section.

Comment: Which is the seeded cell density in the 2D conditions? From the figure 1 it seems that it is very different among the experimental conditions, and it can affect also the expression of the pluripotent and epithelial differentiation markers, because when ESC overgrowth they start to differentiate spontaneously.

 Answer: The seeding density was 1x105 cells / cm2 in all conditions. This has been now added to the material and methods section.

Reviewer 3 Report

This paper aims to identify culture conditions for embryonic stem cells, involving cell priming and differentiation stages, that support epithelial maturation. This is a well written manuscript with promising results. The main area of improvement I see would be in comparing these results to other work. A starting point would be Sakulpaptong et al., 2022 (https://doi.org/10.1371/journal.pone.0263083) and its references. This is interesting work that I think would be more impactful if given more context in the field of tissue engineering. All other suggestions listed below are very minor, largely related to formatting.

 1. Check formatting of subsection titles on page 4-line 133: “2.3.3.”, page 7-line: 172: “2.4.2.”, page 8-line 186 “2.5.2.” and page 10-line 287 “4.33.”

2. Figure 3: it looks like “L” is missing for the image. It might be worth adding to the caption that the scale bars shown in A and B apply for all images below in the columns, or that the right column is an expansion of the left column (same for Figure 4).

3. Page 9, 248: sentence appears incomplete.

4. Page 9, lines 283-4: check formatting of cell seeding density (1x105 cells / cm2) and please mention the cell seeding density used for the 3D cultures; I’m assuming it was also 1x105cells/cm2 but it would be helpful to clarify.

5. Page 9, lines 261 & 264: references should be numbered as in text above.

6. Page 10, line 292: please mention fibroblast seeding density.

Author Response

Thank you for the thorough review of our paper that helped improving it. Please find bellow point by point our answers. We have also revised the English language and we hope that the manuscript is now acceptable for publication.

Comment: This paper aims to identify culture conditions for embryonic stem cells, involving cell priming and differentiation stages, that support epithelial maturation. This is a well written manuscript with promising results. The main area of improvement I see would be in comparing these results to other work. A starting point would be Sakulpaptong et al., 2022 (https://doi.org/10.1371/journal.pone.0263083) and its references. This is interesting work that I think would be more impactful if given more context in the field of tissue engineering.

Answer: We thank to reviewer for the very interesting literature suggestion. We have included a paragraph in the discussion on the relevance of this work in the field of tissue engineering and included the paper of Sakulpaptong et al., 2022 in the reference list.

Formatting comments:  

Comment:  1. Check formatting of subsection titles on page 4-line 133: “2.3.3.”, page 7-line: 172: “2.4.2.”, page 8-line 186 “2.5.2.” and page 10-line 287 “4.33.”

Answer: Thank you. This has been now corrected.

Comment: 2. Figure 3: it looks like “L” is missing for the image. It might be worth adding to the caption that the scale bars shown in A and B apply for all images below in the columns, or that the right column is an expansion of the left column (same for Figure 4).

Answer: Thank you. We have added L to the image and the explanation about the scale bars to the figure legends.

Comment: 3. Page 9, 248: sentence appears incomplete.

Answer: Thank you very much for pointing to us this mistake. It has been now corrected.

Comment: 4. Page 9, lines 283-4: check formatting of cell seeding density (1x105 cells / cm2) and please mention the cell seeding density used for the 3D cultures; I’m assuming it was also 1x105cells/cm2 but it would be helpful to clarify.

 Answer: The seeding density was 1x105 cells / cm2 in all conditions. This has been now added to the material and methods section.

Comment: 5. Page 9, lines 261 & 264: references should be numbered as in text above.

Answer: Thank you very much for pointing to us this mistake. It has been now corrected.

Comment: 6. Page 10, line 292: please mention fibroblast seeding density.

 Answer: The seeding density of fibroblasts in collagen gels was 0.5x106 cells / ml matrix in all conditions. This has been now added to the material and methods section.

Round 2

Reviewer 1 Report

Thank you for taking the time to carefully address the comments, making changes wherever possible. It is my hope that the authors will delve deeper into the molecular mechanism in their follow-up manuscript.

In my view, the manuscript appears favorable after the revisions and is suitable for acceptance in its modified format.

Author Response

Dear reviewer, thank you very much for your comments. We also believe the manuscript has become much better after your suggestions.

Reviewer 2 Report

The authors reported that they seeded 1x105 cells / cm2, but they did not explain how the seed density is so different among the conditons. I think it is a very important point, since the cell density could affect differentation efficiency.

Author Response

Dear reviewer, we agree that from Figure 1 one might get the impression that the seeding density was different. We seeded initially in all conditions the number of cells per surface unit. The images in Figure 1 are 8 days after seeding in different conditions, thus after several changes of the culture medium, so cells that did not attach have been removed, giving maybe the impression of different seeding densities. In this sense the reviewer is right that the actual number of attached cells might have been different between different conditions, but it was difficult for us to adjust the number of seeded cells between conditions such that the number of attached cells would be the same in all conditions, so in order to avoid inducing even more variation by adjusting the number of seeding cells, we kept seeding the ESC cells at the same seeding density on all conditions, either different media formulations or different ECM substrates. 

Round 3

Reviewer 2 Report

The authors addressed all the reviewer's comments